# Genome-wide association analysis and replication in 810,625 individuals with varicose veins

Waheed-Ul-Rahman Ahmed [1,8], Sam Kleeman [2,8], Michael Ng [1], Wei Wang [3], Adam Auton[3], 23andMe Research Team*, Regent Lee [4], Ashok Handa[4], Krina T. Zondervan[5,6], Akira Wiberg [1,7,9] & Dominic Furniss [1,7,9 ✉]

Varicose veins affect one-third of Western society, with a significant subset of patients developing venous ulceration, costing $14.9 billion annually in the USA. Current management consists of either compression stockings, or surgical ablation for more advanced disease. Most varicose veins patients report a positive family history, and heritability is ~17%. We describe the largest two-stage genome-wide association study of varicose veins in 401,656 individuals from UK Biobank, and replication in 408,969 individuals from 23andMe (total 135,514 cases and 675,111 controls). Forty-nine signals at 46 susceptibility loci were discovered. We map 237 genes to these loci, several of which are biologically plausible and tractable to therapeutic targeting. Pathway analysis identified enrichment in extracellular matrix biology, inflammation, (lymph)angiogenesis, vascular smooth muscle cell migration, and apoptosis. Using a polygenic risk score (PRS) derived in an independent cohort, we demonstrate its predictive utility and correlation with varicose veins surgery.

[1] Nuffield Department of Orthopaedics, Rheumatology and Musculoskeletal Sciences, University of Oxford, Botnar Research Centre, Windmill Road, Oxford OX3 7LD, UK. [2] Cold Spring Harbor Laboratory, Cold Spring Harbor, NY 11724, USA. [3] 23andMe, Inc., Sunnyvale, CA, USA. [4] Nuffield Department of Surgical Sciences, University of Oxford, John Radcliffe Hospital, Oxford OX3 9DU, UK. [5] Nuffield Department of Women's & Reproductive Health, University of Oxford, John Radcliffe Hospital, Oxford OX3 9DU, UK. [6] Wellcome Centre for Human Genetics, University of Oxford, Old Road Campus, Roosevelt Drive, Oxford OX3 7BN, UK. [7] Department of Plastic and Reconstructive Surgery, Oxford University Hospitals NHS Foundation Trust, John Radcliffe Hospital, Oxford OX3 9DU, UK. [8] These authors contributed equally: Waheed-Ul-Rahman Ahmed, Sam Kleeman. [9] These authors jointly supervised this work: Akira Wiberg, Dominic Furniss. *A list of authors and their affiliations appears at the end of the paper. ✉email: dominic.furniss@ndorms.ox.ac.uk

Varicose veins (VVs) are a very common manifestation of chronic venous disease, affecting over 30% of the population in Western countries[1]. In the USA, chronic venous disease affects over 11 million males and 22 million females aged 40–80 years[2], meaning it is twice as prevalent as coronary heart disease[3]. Chronic venous insufficiency leads to serious complications in 10% of cases, including lipodermatosclerosis, venous ulceration and rarely amputation. Despite best care, 25–50% of venous leg ulcers remain unhealed after six months of treatment[4]. Ongoing management of venous leg ulcers costs around $14.9 billion annually and 4.5 million work days per year are lost to venous-related illness in the USA[5,6]. Despite this, at present no medical treatments exist for VVs. For symptomatic patients, endovenous ablation is the first-line treatment approach[7]. However, recurrence following surgery is 20%, with no difference in recurrence compared to conventional open surgery[8].

VVs are thought to develop from a combination of valvular insufficiency, venous wall alterations, and haemodynamic changes that precipitate venous reflux, stasis and hypertension of the venous network, causing varicosities[9,10]. Risk factors for VVs include older age, female sex, pregnancy, a positive family history, obesity, tall height and previous deep-vein thrombosis (DVT)[3,11,12]. Many patients with VVs report a positive family history[13], and among offspring with one affected parent the familial standardised incidence ratio is 2.39[14], with a heritability of 17%[15], suggesting a genetic component to aetiology. Two recent genome-wide association studies (GWAS) of VVs have been described. Ellinghaus et al.[16] tested for associations in 323 cases and 4619 controls, with suggestive associations examined in an independent cohort totalling 1946 cases and 3146 controls. They reported two associations, mapped to *EFEMP1* and *KCNH8*. Fukaya et al.[17] used UK Biobank to identify a further 30 putative associations associated with VVs. However, their cases were defined only by the International Classification of Diseases (ICD) diagnostic codes, meaning that thousands of cases defined by operation codes were misclassified as controls[18]. Moreover, the genetic associations discovered were not replicated in an independent cohort.

To advance substantially our understanding of the aetiology and genetic architecture of VVs, we performed the largest genome-wide association study (GWAS) of surgically confirmed VVs. We use data from participants from the UK Biobank ($n = 401,656$) and confirm top associations in a large independent cohort of research participants with self-reported VVs from 23andMe ($n = 408,969$). We examine patterns of expression of genes in clinically relevant biologic pathways and prioritise targets for therapeutic development. Furthermore, we derive a polygenic risk score (PRS) for VVs in an independent cohort (FinnGen), and demonstrate its predictive utility in UK Biobank.

## Results
The overall analytic workflow is shown in Fig. 1.

**Association analysis**. Genome-wide testing of the UK Biobank discovery cohort (22,473 cases and 379,183 controls) yielded genome-wide significant associations ($P < 5 \times 10^{-8}$) at 108 risk loci (Supplementary Data 1). Conditional regression yielded a further seven independent signals at six of the 108 loci associated significantly with VVs. As expected, the large discovery sample led to the $\lambda_{GC}$ showing inflation (1.25) (Supplementary Fig. 1). The linkage equilibrium score regression (LDSC) intercept (1.06), and attenuation ratio of 0.13 is consistent with polygenicity[19]. We estimated the total SNP heritability ($h^2_g$) for VVs in UK Biobank to be 8.04% (S.E. = 0.17%) using a variance components method, and 5.03% (S.E. = 0.30%) using LDSC[19]. The estimated

heritability in the 23andMe dataset (113,041 cases and 295,928 controls) was 5.40% (S.E. = 0.30%).

We tested the top 116 signals at the 108 risk loci in the independent 23andMe dataset. Of the 116 associated variants, 106 passed QC and were available in the 23andMe summary statistics. Forty-nine of 116 variants demonstrated significant association at a conservative Bonferroni-corrected threshold of $P < 4.72 \times 10^{-4}$. Thus, we identified 49 significant associations at 46 independent risk loci (Fig. 2; regional association plots for all 49 signals can be found in Supplementary Fig. 2). Allelic effects were concordant across both cohorts at all 49 replicated variants, with minimal evidence of heterogeneity between the two GWAS at all loci (Q-statistic > 0.05). Eighteen loci were previously reported (Supplementary Data 2). Sixty-seven variants at 67 loci did not replicate (Supplementary Data 3); however, there was high overall concordance in the effect sizes of the 106 SNPs between the two datasets (Pearson's $r^2 = 0.84$, $P < 1 \times 10^{-16}$; Supplementary Fig. 3A).

**In silico annotation**. To interrogate and annotate SNPs at our susceptibility loci, we used FUMA (Functional Mapping and Annotation of GWAS)[20]. FUMA identified 5315 genome-wide significant candidate SNPs from our discovery cohort associated with VVs at 45 of the 46 replicated loci (Supplementary Fig. 4; Supplementary Data 4). Around 2% of candidate SNPs ($n = 103$) were exonic, of which 56 were non-synonymous (52 missense, two stop-gain, one splice site variant, and one frameshift variant). Of the non-synonymous variants, four missense variants were predicted to affect protein structure or function, and were in moderate linkage disequilibrium (LD) ($r^2 \geq 0.22$ and $D' \geq 0.75$) with the index SNP at three loci—12q13.12 (index SNP: rs7308356), 16q24.3 (index SNP: rs2002833) and 17q24.3 (index SNP: rs9895127) (Supplementary Data 5). This includes rs7184427 (A/G) ($P = 9.10 \times 10^{-40}$, OR = 1.19, $r^2_{index} = 0.22$, $D'_{index} = 0.75$), which causes a predicted deleterious p.Val250Ala substitution within *PIEZO1* (SIFT[21] score: 0). *PIEZO1* encodes a mechanically active ion channel involved in the detection of vascular shear stress, and was previously associated with VVs and lymphoedema[17,22].

Of the 4294 intronic and intergenic variants identified by FUMA (Supplementary Fig. 4), 3735 (87.0%) resided in open chromatin regions (Supplementary Data 4), and 163 demonstrated evidence of functionality with CADD (Combined Annotation-Dependent Depletion[23]) score ≥12.37, the threshold suggested for deleterious variants (Supplementary Data 6). Using RegulomeDB (RDB[24]) to investigate their regulatory functions, 17 had a RDB score of at least 2b (*likely to affect binding*) and eight had an RDB score of at least 1f (*likely to affect binding and linked to expression of a gene target*).

To narrow down the list of probable functional variants, we performed functionally informed fine-mapping[25] for each replicated locus. Eighteen SNPs were identified with posterior probability >95% (Supplementary Table 1), including two likely functional variants in *PIEZO1* (rs112070238 and rs8053350), affecting the 5′UTR and promoter regions, respectively.

**Gene mapping**. Positional mapping in FUMA SNP2GENE highlighted 204 genes based on genomic position at 38 loci (Supplementary Data 7)[20]. eQTL mapping, based on GTEx v8 tibial artery tissue[26], mapped a total of 80 genes. Genome-wide, gene-based association analysis implemented in MAGMA v1.07[27] identified 248 protein-coding genes significantly associated with VVs ($P < 2.67 \times 10^{-6}$); 117 were within our replicated loci (Supplementary Data 8; Supplementary Fig. 5).

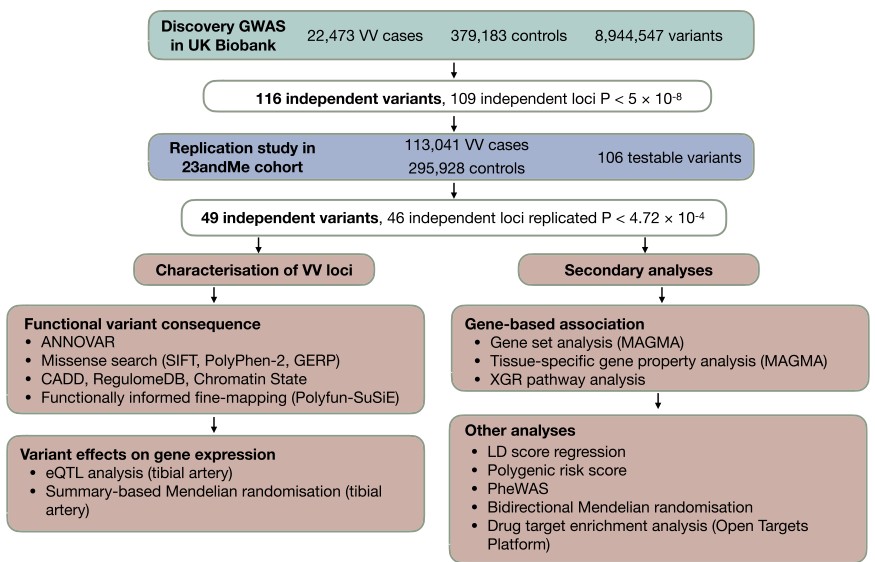

**Fig. 1 Study design and analysis workflow.** A discovery GWAS was performed in the UK Biobank cohort, with the top independent lead variants tested within the 23andMe replication cohort. Forty-nine independent variants at 46 loci met the Bonferroni-corrected threshold in the replication cohort, and were interrogated further in subsequent analyses.

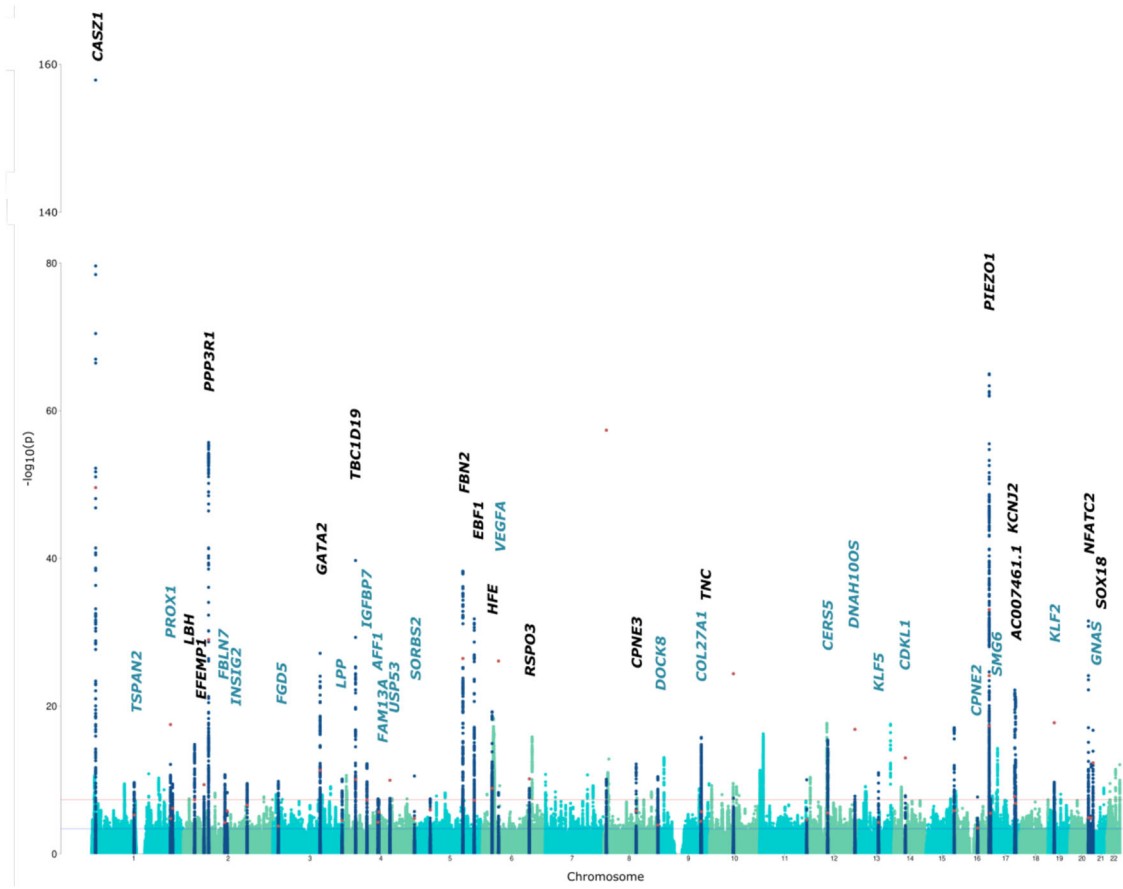

**Fig. 2 Results of genome-wide association study in varicose veins.** Manhattan plot showing genome-wide $P$-values plotted against position on each of the autosomes. The dark blue, light blue and green dots refer to the discovery UK Biobank Cohort, with the red dots corresponding to the 49 variants from the 23andMe cohort at each replicated locus (shown in Supplementary Data 2). The dark blue peaks correspond to the 46 loci that replicated in the 23andMe cohort at a Bonferroni-corrected threshold of $P < 4.72 \times 10^{-4}$. Candidate genes at each locus are named above each signal, with previously unreported genetic loci in blue, and previously described loci in black.

In the summary-based Mendelian randomisation (SMR) analysis[28], testing was performed for 4946 probes with a cis-eQTL at $P < 5 \times 10^{-8}$; a threshold of significance was set at $P_{SMR} < 1.01 \times 10^{-5}$ (0.05/4,946). Forty-four genes passed the set significance threshold ($P_{SMR} < 1.01 \times 10^{-5}$). To exclude SMR associations due to linkage, we performed HEIDI analysis across 44 significant genes −25 passed the HEIDI test ($P_{HEIDI} \geq 1.12 \times 10^{-3}$; Supplementary Data 9), 12 of which were within our VVs susceptibility loci, highlighting an association with VVs through pleiotropy rather than LD and co-localisation.

In summary, 237 unique genes were mapped by at least one mapping approach to 39 of the 46 susceptibility loci. Significant overlap between the mapping strategies was seen, with the majority of genes (54.9%, $n = 130$) being prioritised by two or more mapping approaches. Thirty-six genes were prioritised by three mapping approaches, and six genes (ATF1, AP1M1, DNAH10OS, FBLN7, LBH, WDR92) were prioritised by all four approaches.

**Gene set, tissue-specific and pathway enrichment**. To delineate gene sets and enriched pathways where the 237 prioritised genes converged, gene-set analysis was conducted in MAGMA v1.07[27]. Following MAGMA gene-set enrichment analysis, four Gene Ontology (GO) gene sets were significantly over-represented in our data: cardiovascular development ($P = 1.56 \times 10^{-8}$, $n = 666$); tube morphogenesis ($P = 9.35 \times 10^{-8}$, $n = 778$); blood vessel morphogenesis ($P = 9.39 \times 10^{-7}$, $n = 555$); and tube development ($P = 1.68 \times 10^{-6}$, $n = 956$) (Supplementary Table 2). Further, tissue-specific gene property analysis demonstrated significant gene expression in all three vascular tissue types present in GTEx 54 tissue types: coronary artery ($P = 6.23 \times 10^{-7}$, 2nd most enriched), tibial artery ($P = 1.05 \times 10^{-6}$, 3rd most enriched) and aorta ($P = 3.92 \times 10^{-5}$, 8th most enriched; Supplementary Fig. 6). MAGMA analysis of GTEx 30 general tissue types also demonstrated blood vessels to be a highly enriched tissue ($P = 3.8 \times 10^{-4}$, 3rd most enriched). Using eXploring Genomic Relations (XGR) analysis[29], six canonical pathways were significantly enriched. This included enrichment for genes in pathways related to extracellular matrix biology, the VEGF and VEGFR signalling network, and intracellular $Ca^{2+}$ signalling in the T-cell receptor (TCR) Pathway (Table 1).

**Polygenic risk score analyses**. In order to derive a polygenic risk score (PRS) without overfitting to the UKB cohort, we utilised VV summary statistics from the FinnGen cohort (17,027 cases, 190,028 controls), which were highly concordant with the UK Biobank GWAS (Pearson's $r^2 = 0.95$, $P < 1 \times 10^{-16}$; Supplementary Fig. 3B). Using multivariable logistic regression, we found that the FinnGen-derived PRS was an independent predictor for

varicose vein case/control status (Fig. 3 and Supplementary Table 3). Individuals in the top PRS decile had markedly increased odds of VV diagnosis (OR 4.57, 95% CI 4.27–4.92, $P < 1 \times 10^{-300}$) compared to the bottom PRS decile. We hypothesised that VV cases who underwent surgery are phenotypically more severe and so would have a greater genetic liability than non-surgical VV cases. In a subgroup analysis of UK Biobank VV cases defined by self-reported diagnosis, ICD-10 or OPCS-4 code, history of surgery was associated with significantly increased PRS (0.407 vs. 0.330; $P = 5.5 \times 10^{-4}$), suggesting that the VV-PRS captures disease severity.

**Phenome-wide VV associations**. To uncover associations between VV and other phenotypes, we first performed genetic correlation analysis using LD score regression[19]. Of the 176 traits from nine categories tested for genetic correlation, twelve traits from two categories (anthropometric and autoimmune) met our Bonferroni-corrected significance threshold ($P < 5.56 \times 10^{-3}$; Supplementary Table 4). All twelve significant traits were positively correlated with VVs ($r_g$ range: 9–21%). Eleven traits belonged to the anthropometric category and pertained to height and weight phenotypes, which are established risk factors for VVs[3,11,12].

In the autoimmune trait category, we discovered systemic lupus erythematosus (SLE) to be correlated with VVs, sharing ~19% genetic overlap ($P = 4.2 \times 10^{-3}$, $r_g = 0.195$). Of note, the C allele of variant rs17321999 at 2p23.1 (LBH), which is associated with an increased risk of SLE ($P = 2.22 \times 10^{-16}$, OR = 1.20)[30], was also significantly associated with VVs in our discovery cohort ($P_{disc} = 3.20 \times 10^{-14}$, OR = 1.09), and in high LD with lead SNP at 2p23.1 in the meta-analysis (rs9967884; $P_{meta} = 1.40 \times 10^{-14}$, OR = 1.11; $r^2 = 0.87$). We used the Open Targets Genetics platform[31] to confirm the associations between VVs and SLE at the variant level, using a phenome-wide association study (PheWAS) approach (Supplementary Data 10); three of our 49 replicated lead variants (rs4849044, rs7773004, rs61863928) demonstrated an association with SLE.

As a complementary approach, we leveraged the FinnGen-derived VV-PRS to perform a phenome-wide polygenic risk score analysis using a set of curated time-to-event phenotypes ($n = 694$) in UK Biobank. Eighteen phenotypes met the phenome-wide significance threshold ($P = 1 \times 10^{-5}$) in the Cox regression (Supplementary Table 5). The most significantly associated phenotypes were "Phlebitis and thrombophlebitis", "Unspecified monoarthritis", "Obesity", "Postphlebitic syndrome", "Pulmonary embolism and infarction" and "Umbilical hernia". To investigate whether these phenotypes are causally associated with VV, we performed bidirectional Mendelian randomisation using genetic instruments derived from our UK Biobank GWAS. Using the inverse-variance weighted (IVW)

**Table 1 Gene-based enrichment analysis.**

| Biological process | Z-score | P-value | FDR | Number of overlapped genes | Genes |
|---|---|---|---|---|---|
| Alpha9 beta1 integrin signalling events | 3.85 | 0.0012 | 0.032 | 2 | TNC, VEGFA |
| Genes encoding structural ECM glycoproteins | 3.39 | 0.0012 | 0.032 | 6 | EFEMP1, FBLN7, FBN2, IGFBP7, RSPO3, TNC |
| Calcium signalling in the CD4 + TCR pathway | 3.5 | 0.0019 | 0.032 | 2 | NFATC2, PPP3R1 |
| Ensemble of genes encoding core extracellular matrix, including ECM glycoproteins, collagens and proteoglycans | 3.11 | 0.0019 | 0.032 | 7 | COL27A1, EFEMP1, FBLN7, FBN2, IGFBP7, RSPO3, TNC |
| Non-canonical WNT signalling pathway | 3.29 | 0.0025 | 0.035 | 2 | MAPK10, NFATC2 |
| VEGF and VEGFR signalling network | 3.11 | 0.0032 | 0.036 | 1 | VEGFA |

Ontology enrichment analysis was performed across all mapped genes using XGR, in "canonical pathways", with the following settings: hypergeometric distribution testing, any number of genes annotated, any overlap with input genes and an adjusted false-discovery rate (FDR) < 0.05. The Z-scores, P-values, false-discovery rate and the overlapped genes for each of the ontologies are shown.

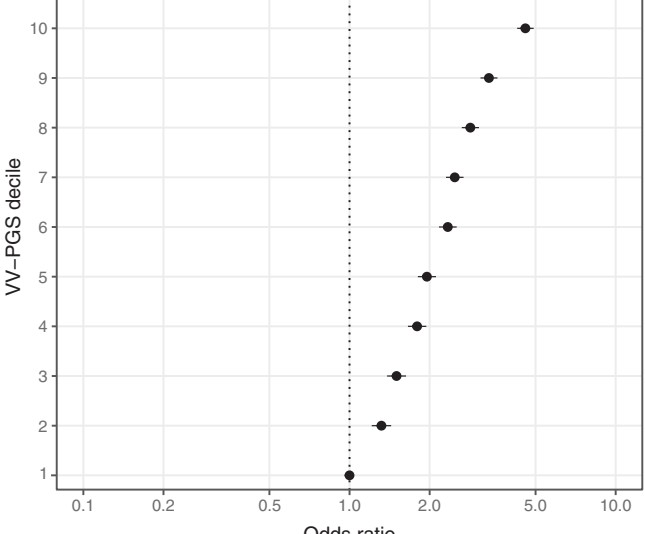

**Fig. 3 Logistic regression for varicose vein case/control status in UK Biobank against VV-polygenic risk score (VV-PRS) decile.** The VV-PRS was derived in an independent study sample, FinnGen ($n = 17,027$ VV cases and 190,028 controls). Odds ratio refers to odds ratio of VV in UK Biobank versus decile 1 of VV-PRS (lowest). Error bars signify 95% confidence intervals of the odds ratios.

method as the primary analysis, we found evidence for raised body mass index being causal in VV, with no evidence for significant directional pleiotropy on MR-Egger sensitivity analysis (Supplementary Table 6). Two further suggestive causal associations were found: VVs (exposure phenotype) on thrombophlebitis risk and hernia (exposure phenotype) on VVs risk.

**Drug-target enrichment analysis.** Finally, we tested the amenability of our associated genes to drug targeting using the Open Targets Platform[32]. Of the 237 mapped genes, 200 genetic targets were identified[32]. Forty-two drug pathways reached nominal significance ($P < 0.05$), with the Butyrophilin family interactions pathway being most enriched ($P = 2.6 \times 10^{-7}$, six targets), followed by the Calcineurin-NFAT pathway ($P = 9.4 \times 10^{-4}$, two targets), and the Transcriptional regulation by RUNX1 pathway, which possessed the highest number of targets ($P = 1.6 \times 10^{-3}$, 14 targets; Supplementary Data 11). Tractability information for 105 gene targets was available, with 65 of 237 genes predicted to be tractable to antibody targeting to a high confidence, and 26 genes predicted to be tractable to small molecule targeting (Supplementary Data 12). Eight of the 237 genes overlapped with pharmacologically active targets with known pharmaceutical interactions (*CDK10, COL27A1, GABBR1, KCNJ2, MAPK10, OPRL1, TNC* and *VEGFA*) (Supplementary Data 13). Of note, *VEGFA* is a target for several antibody, protein and oligosaccharide agents, which are currently in phase 2, 3 and 4 clinical trials, including for several ocular vascular disorders.

**Discussion**
VVs cause significant morbidity and large healthcare costs. There is a compelling need for a deeper understanding of the biology of VVs in order to develop new therapeutic strategies. Our analysis represents the largest and most comprehensive GWAS to date of VVs, including 135,514 patients and 675,111 controls. We discovered 28 previously unreported risk loci (29 signals), and independently replicated 18 of 29 previously reported but non-replicated loci (20 of 32 known signals). Our in silico analyses demonstrated strong evidence of functional variants in VV-

associated genomic regions. Furthermore, our pathway analyses establish a strong enrichment for genes expressed in the extracellular matrix, immune cell signalling and circulatory system development. We have also identified a previously unreported genetic correlation between VVs and systemic lupus erythematosus (SLE). Several prioritised genes demonstrate the potential for pharmacological targeting, and are currently under active investigation in other diseases. Lastly, we demonstrate the potential utility of a polygenic risk score (PRS) in VVs, demonstrating that genetic risk correlates with a more severe phenotype —a first step in facilitating personalised medicine approaches to management.

**Biological drug targets.** This study has identified several biological pathways as mediators of VV pathophysiology that may be of relevance to pharmacological targeting. These cluster broadly into five functional categories: extracellular matrix regulation, the immune response, angiogenesis and lymphangiogenesis, smooth muscle cell biology and apoptosis (Supplementary Data 14), and we will briefly examine the three functional categories that produced the most compelling evidence in this study.

**Extracellular matrix regulation.** VVs demonstrate an increased luminal diameter and intimal hypertrophy, features closely related to disruption of the extracellular matrix (ECM)[33]. Deposition of ECM in the perivascular space in VVs is also recognised— possibly a compensatory mechanism to reinforce an already weakened wall[34]. Venous dilatation and valve ring enlargement seen in VVs is thought to affect the ability of venous valves to coapt, which in turn contributes to venous reflux and hypertension[10]. Indeed, there is a noted imbalance of ECM proteins in VVs, specifically collagen and elastin—with a preponderance of collagen compared to normal vein[35]. It is therefore possible that disruption in intrinsic connective tissue components of the vein wall or valves may contribute to VVs.

Our prioritised genes significantly overlapped with canonical pathways relating to ECM components, including Collagen Type XXVII Alpha 1 Chain (*COL27A1*) and EGF-containing Fibulin-like Extracellular Matrix Protein 1 (*EFEMP1*).

rs753085 ($P = 2.17 \times 10^{-11}$, OR = 1.07) is in an intron of *COL27A1*. COL27A1 is a fibrillar collagen in the extracellular matrices of several tissues, including blood vessels[36]. *COL27A1* expression has been demonstrated to be reduced in VV samples[37]. Moreover, our drug-enrichment analysis demonstrated COL27A1 to be a pharmacologically active target, with pharmaceutical agents currently being investigated in several clinical trials (Supplementary Data 13), demonstrating its potential candidacy as a therapeutic target for VV prevention or treatment.

In our GWAS, we replicated the previously described association between rs3791679 and VVs ($P = 1.59 \times 10^{-13}$, OR = 1.08)[16]. rs3791679 resides in an enhancer region of *EFEMP1*, and is believed to be the causal variant at this locus[38]. *EFEMP1* encodes the ECM glycoprotein fibulin-3[39], which plays a key role in maintaining the integrity of elastic tissues[16], and is highly expressed in vein endothelial cells. Interestingly, the same A allele of rs3791679 is also associated with abdominal hernia susceptibility[40]; this pleiotropy may in part explain our Mendelian randomisation finding that placed hernias on the causal pathway for VVs.

Fibulin-3 has been found to antagonise vascular development by reducing the expression of the matrix metalloproteases, MMP2 and MMP3, and increasing expression of tissue inhibitors of metalloproteases (TIMPs) in endothelial cells[41]. The saphenofemoral junction in VVs demonstrates reduced expression of MMP2, and heightened expression of TIMP1 and MMP1 protein levels[42]. Alterations in expression of these enzymes may therefore

result in weakness in the venous wall and predispose patients to VVs. Our drug-enrichment analysis found fibulin-3 to be tractable to antibody targeting to a high confidence; moreover, metformin has been demonstrated to downregulate fibulin-3 through downregulation of MMP2[43]. Fibulin-3 therefore represents another potential pharmacological target for VVs.

**The immune response**. Chronic Inflammation in the venous wall has been proposed to play a role in VV aetiopathology[33]. When compared to normal veins, enhanced expression of inflammatory mediators has been observed[10]. VVs contain increased mast cells, monocytes and macrophages compared to normal veins[10].

We defined five inflammation-associated risk loci in our GWAS. Of particular note is rs78216177 ($P_{meta} = 5.80 \times 10^{-14}$, OR = 1.10), in an intron of *DOCK8* that has roles in both innate and adaptive immune systems. Deletion of *DOCK8* is strongly associated with Hyper-IgE syndrome, a type of primary immunodeficiency that affects multiple systems, including the vasculature[44]. Vascular abnormalities in hyper-IgE syndrome include aneurysmal changes and abnormalities in great vessels.

Notably, we also discovered a significant genetic overlap between VVs and SLE, an autoimmune disease, in the genetic correlation analyses, albeit not in the phenome-wide analysis of VV-polygenic risk score. The associations between VVs and SLE were confirmed at the variant level using a PheWAS approach, with three of 49 replicated lead variants (rs4849044, rs7773004, rs61863928) demonstrating an association with SLE. Furthermore, lead variant rs1471251 ($P = 8.33 \times 10^{-11}$, OR = 1.06) is a known eQTL of *AFF1*, which is associated with SLE[45]. Reinforcing this shared genetic basis, a VVs-associated variant at *LBH*, rs17321999 ($P_{disc} = 3.20 \times 10^{-14}$, OR = 1.09), was also previously associated with SLE ($P = 2.22 \times 10^{-16}$, OR = 1.20)[30].

XGR analysis demonstrated enrichment of "intracellular calcium signalling in the CD4 + T-cell receptor (TCR) pathway" ($P = 1.9 \times 10^{-3}$, Z = 3.5), specifically highlighting genes *NFATC2* and *PPP3R1*, which are intimately involved in this pathway. rs3787184 ($P_{meta} = 2.51 \times 10^{-36}$, OR = 1.16) was one of 18 SNPs identified as likely causal variants in our functionally informed fine-mapping (Supplementary Table 1), and resides in a promoter region of *NFATC2*. rs2861819 ($P_{meta} = 2.65 \times 10^{-77}$, OR = 1.20) is in an intergenic region ~19 kb upstream of *PPP3R1*. *PPP3R1* encodes calcium binding B (CnB), a subunit of calcineurin, a $Ca^{2+}$influx-activated serine/threonine-specific phosphatase that interacts with NFAT transcription factors in the regulation of naive T-cell activation[46]. VVs are defined by clustering and infiltration of T lymphocytes[34], which are predominantly distributed close to the venous valve agger[47], a fibroelastic structure located at the base of venous valves where tunica media meets adventitia. Therefore, we hypothesise that aberrant *PPP3R1* and *NFATC2* expression could alter calcium signalling in T-cells, which may contribute to the valvular pathology seen in VVs. The Calcineurin-NFAT pathway was also the second-most enriched in our drug-target enrichment analysis, and may represent a novel therapeutic approach to VVs.

**Angiogenesis and lymphangiogenesis**. Disruption in normal angiogenic processes can lead to VVs[48], potentially because of a failure to develop properly formed venous walls and valves, or to repair defects following vascular stress injury. MAGMA gene-set analysis revealed enrichment of several gene sets relating to tube formation and morphology, with the VEGF/VEGFR signalling network also being enriched in the XGR analysis.

rs11967262 at 6p21.1 ($P_{meta} = 1.45 \times 10^{-19}$, OR = 1.09) lies in an intergenic region ~7 kb upstream of vascular endothelial growth factor A (*VEGFA*). VEGFA is a critical regulator of angiogenesis, and is fundamental to maintaining the integrity and functionality of the vessel wall[49]. VEGFA is a selective endothelial mitogen, binding to its receptor VEGFR2 to induce endothelial cell proliferation, migration and differentiation. VEGFA and VEGFR2 expression are significantly enhanced in the wall of VVs compared to normal veins, particularly in VVs complicated by thrombophlebitis[50]; interestingly, our Mendelian randomisation analysis found evidence for a causal role played by VVs in thrombophlebitis risk. Plasma levels of VEGFA have also been demonstrated to be significantly increased in patients with VVs[51]. VEGFA also causes vasodilatation, which reduces vessel tone, leading to stasis and the release of oxygen free radicals, which contributes to vein wall weakness[52]. Intriguingly, VEGFA also promotes inflammation via expression of intercellular and vascular cell adhesion molecules, linking angiogenesis to immune dysregulation[53]. Of note, anti-VEGFA agents are currently being investigated in several clinical trials for the treatment of retinal vein occlusion (Supplementary Data 13). The VEGF axis therefore represents a promising candidate for therapeutic targeting in the treatment of VVs.

Positional and MAGMA mapping highlighted genes at four loci relating to lymphangiogenesis. Indeed, the lymphatic system develops from veins, and its function is intimately related with the venous circulation, draining extracellular fluid back into circulation. It is therefore feasible that similar genetic defects may result in either lymphoedema, varicose veins or a combination of the two conditions. *PROX1* is a master inducer gene necessary for the development of lymphatic vasculature[54], and *PROX1*knock-out mice are deficient of lymphatic vasculature[55]. During developmental lymphangiogenesis, *PROX1* has been shown to be necessary for the formation of lymphovenous valves[56,57], suggesting it may predispose to VV development by causing defects in venous valves. rs340875, ($P_{meta} = 4.22 \times 10^{-20}$, OR = 1.09) is ~2 kb upstream of *PROX1*, and we also fine-mapped two probable functional variants at this locus (Supplementary Table 1).

Furthermore, *PROX1* is co-expressed and functions alongside the transcription factor *FOXC2* in lymphatic valve-forming cells at the earliest stage of lymphatic development[58]. Mutations in *FOXC2* cause hereditary lymphoedema-distichiasis—a disease characterised by VVs and peripheral lymphoedema—highlighting genetic overlap between the two disorders[59]. A previous analysis also suggested *FOXC2* to be implicated in the development of varicose veins in the general population[60], but we did not find evidence of association between varicose veins and *FOXC2* in our study. In fact, the only gene identified in this study known to be associated with a Mendelian disease that results in VVs is *PIEZO1*, which also predisposes to both VVs and lymphoedema (Supplementary Table 7).

**Polygenic risk score**. Over two million people in the USA have advanced chronic venous disease[2], and around 500,000 per year undergo invasive surgical procedures. Our polygenic risk score (PRS) analysis of operated vs unoperated VVs cases represents an important proof-of-principle, demonstrating the feasibility of enhanced prognostication in enabling the subset of VVs patients that will require surgical intervention to be identified. Preventive strategies in high-risk individuals could include prophylactic compression stocking use, or early ablation procedures to mitigate the risk of venous ulceration. The benefits of early endovenous ablation in improving healing of venous leg ulcers has been demonstrated[61]. Future research may also enable the identification of those at high risk of recurrence following surgery, a significant problem in current management of VVs patients.

**Strengths and limitations**. Certain limitations of this study must be acknowledged. While the discovery GWAS in UK Biobank used a combination of hospital diagnostic codes, operation codes

and self-report codes, VV cases in 23andMe were identified based on self-report alone, meaning that the phenotyping for the replication GWAS was necessarily less stringent. Moreover, rather than undertake a formal meta-analysis between the discovery and replication GWAS, we independently tested the association for the 116 independent lead SNPs that were genome-wide significant in the UK Biobank discovery GWAS. Thus, sub-threshold signals in the discovery GWAS that may have reached genome-wide significance in the replication GWAS, or under meta-analysis, were not identified. Furthermore, not having access to the full summary statistics for the replication GWAS also meant that our in silico analyses were performed on the summary statistics from the discovery GWAS alone.

Finally, we acknowledge that out of the many VVs-associated genes identified by a multitude of gene-prioritisation and mapping methods, only a small minority have robust evidence linking them to VVs through a truly causal variant. Moreover, our eQTL-based gene mapping and summary-based Mendelian randomisation analyses were limited by our lack of access to eQTL data for venous tissue. We therefore had to employ GTEx tibial artery tissue data as an imperfect surrogate, reasoning that it is the anatomically, histologically, and embryologically closest tissue to lower limb veins within the GTEx dataset. Functional validation will be required in diseased venous tissue to confidently associate these candidate susceptibility variants and genes to VVs.

Several strengths of the paper mitigate these limitations. We have performed the largest GWAS of VVs to date by a considerable margin, with 135,514 cases and 675,111 controls. We have stringently controlled our false-positive rate by reporting only the loci that were genome-wide significant in the discovery GWAS *and* that subsequently replicated, so we can be confident in the veracity of these 49 signals. This is reflected in the plethora of biologically plausible genes, gene clusters and biological pathways that were associated with these loci. Furthermore, by including surgical codes for phenotyping in the UK Biobank discovery GWAS, we were able to identify a considerably greater number of cases than a previous GWAS that also used the UK Biobank resource but relied on ICD diagnostic codes alone (22,473 vs. 9577 cases)[17,18]. As a general principle of case ascertainment, we believe there is much to be gained by seeking out individuals who have undergone surgery for a disease: given the inevitable risk of complications, surgery is generally reserved for those at the more phenotypically severe end of the spectrum who have failed non-surgical treatment. Finally, the predictive capabilities of a VVs polygenic risk score derived in an independent cohort not only underscores the validity of the VVs-associated polygenic signals uncovered in our discovery GWAS, but also opens the door to the future use of genetic risk-stratification in improving prognostication and guiding decision-making in the management of VVs patients.

In summary, we have described the largest GWAS to date of VVs, a highly prevalent disease with a substantial health and socio-economic cost. We discovered 49 variants at 46 loci that predispose to VVs. We have identified pathways and genes involved in extracellular matrix regulation, inflammation, vascular and lymphatic development, smooth muscle cell activity and apoptosis, all of which are biologically plausible contributors to the pathobiology of VVs, and provide excellent candidates for further investigation of venous biology. Several genes appear tractable to pharmacological targeting (notably *VEGFA*, *COL27A1*, *EFEMP1*, *PPP3R1* and *NFATC2*), and may represent viable therapeutic targets in the future management of VV patients. Importantly, our polygenic risk score represents a first step towards better prognostication in patients with VVs.

## Methods
**Ethical approval and consent**. UK Biobank obtained ethical approval from the North West Multi-Centre Research Ethics Committee (MREC) (11/NW/0382) to collect and disseminate data and samples from participants (for more details: ukbiobank.ac.uk/ethics). This study was conducted under UK Biobank study ID 22572. All participants provided informed consent for their genotype data to be used for this research. The consent procedures for UK Biobank are provided elsewhere (for more details: www.ukbiobank.ac.uk/). All 23andMe participants were obtained from the customer base of 23andMe, Inc. Genotyping of participants was performed by the 23andMe Personal Genome Service. All 23andMe research participants provided informed consent for their genotype data to be utilised for research purposes under a protocol approved by the external AAHRPP-accredited IRB, Ethical & Independent Review Services. Further details regarding the consent processes of 23andMe can be found elsewhere (23andme.com/en-gb/about/consent).

**Population and phenotype definition**. The UK Biobank is a population-level resource comprising a prospective cohort of approximately 500,000 participants, recruited at age 40–69 years, at 22 centres across England, Scotland and Wales, between 2006 and 2010[62]. Participants underwent whole-genome genotyping and data linkage with their medical records was performed to permit deep phenotyping[63]. The characteristics of the full UK Biobank cohort are described in detail elsewhere[64]. In the discovery analysis, VV cases were identified from the UK Biobank data showcase (ukbiobank.ac.uk) using the following diagnostic, operative and self-report codes (Supplementary Table 8):

1. Primary and/or secondary ICD-10 codes for varicose veins (I83)
2. Primary and/or secondary OPCS code for varicose vein surgery: ((L84-L88)
3. Self-reported operation code for varicose vein surgery (1479)
4. Self-reported non-cancer illness code for varicose veins (1494)

In summary, 27,165 individuals from the UK Biobank cohort possessed at least one of the above codes and were classified as VVs cases. Following quality control (QC), we identified 22,473 VV cases and the remaining 379,183 individuals were defined as controls.

In the 23andMe replication cohort, participants answered the question "*Do you have varicose veins on your legs? (Yes/Not Sure/No)*". VV cases were identified if they answered "*Yes*", while controls were those who answered "*No*". In all, 113,041 VV cases and 295,928 controls were included in the final replication analysis.

**Genotyping**. UK Biobank participants were genotyped on UK BiLEVE (49,950 participants) and UK Biobank Axiom arrays (438,427 participants), with ~95% shared content. In all, 805,426 directly genotyped variants from 488,377 participants were available prior to QC. The 23andMe replication cohort was genotyped on one of four custom arrays (v1/v2, v3, v4, v5). Illumina HumanHap550+ BeadChip was used for v1/v2 (1680 cases, 4882 controls) and the Illumina OmniExpress+ BeadChip was used for v3 (21,342 cases, 56,448 controls). For v4 a fully customised array (58,883 cases, 148,637 controls) was used, and for v5, the Illumina Infinium Global Screening Array was used (31,136 cases, 85,961 controls). Successive arrays contained significant overlap with all previous arrays.

**Quality control**. Quality control (QC) for the UK Biobank discovery cohort was conducted using PLINK v1.9 (https://zzz.bwh.harvard.edu/plink/) and R v3.3.1, as previously described (Supplementary Fig. 7)[66]. Briefly, all SNPs with a call rate <90% were removed. Following which, sample-level QC was conducted—individuals were excluded if: (i) they demonstrated heterozygosity >3 S.D. from the mean (calculated using UK Biobank's PCA-adjusted heterozygosity values, Data Field 20004); (ii) there was disparity between genetically inferred sex (Data Field 22001) and self-reported sex (Data Field 31) or individuals with aneuploidy of sex chromosomes (Data Field 22019); and (iii) had a call rate <98%. Further, we removed all individuals who were not of white British ancestry (based on PCA and self-reported ethnicity (Data Field 22006).

To confirm individuals were of white British-ancestry, we combined our UK Biobank data with individuals identified as GBR from the 1000 Genomes Project (www.1000genomes.org), and using FlashPCA v2.0 (github.com/gabraham/flashpca/) we identified agreement between the two datasets. Following this analysis, 86,693 individuals were excluded from our discovery GWAS. Using a linear mixed model implemented in BOLT-LMM[67] allowed us to include and account for related individuals. Next, conducting SNP-level QC, we excluded 230,562 SNPs based on: (i) a call rate <98%, (ii) Hardy–Weinberg Equilibrium (HWE) $P < 1 \times 10^{-4}$, (iii) MAF < 1%. We further excluded six individuals who were visual outliers when autosomal heterozygosity was plotted against call rate. Our post-QC discovery GWAS, therefore, consisted of 401,667 individuals and 547,011 genotyped SNPs (Supplementary Fig. 7). Following QC, our final discovery GWAS consisted of 401,667 individuals of white British ancestry and 547,011 genotyped single-nucleotide polymorphisms (SNPs).

For the 23andMe replication analysis, samples were restricted to individuals from European ancestry determined through an analysis of local ancestry[68]. A maximal set of unrelated individuals was chosen for each analysis using a segmental identity-by-descent (IBD) estimation algorithm, defining related individuals if they shared more than 700 cM IBD, including regions where the two individuals share either one or both genomic segments IBD. Cases were preferentially retained in the analysis. Variant QC was applied independently to

genotyped and imputed GWAS results. The SNPs failing QC were flagged based on multiple criteria, such as HWE *P*-value, call rate, imputation R-square and test statistics of batch effects.

**Imputation**. The phasing and imputation of UK Biobank has been previously described[63]. Briefly, using SHAPEIT3[69] (jmarchini.org/shapeit3/), autosome phasing was conducted, with a reference panel based on Phase 3 1000 Genomes Project data[70]. Imputation was performed using a combined HRC reference panel (www.haplotype-reference-consortium.org/) and a merged UK10000/1000 Genomes Phase 3 Panel[71]. This resulted in an imputation file consisting of 92,693,895 autosomal SNPs, short indels and large structural variants[63]. In 23andMe, out-of-sample modified versions of the Beagle graph-based haplotype phasing algorithm[72] and Eagle2 v2.3[73] algorithm were used to phase samples. We used a 23andMe internally developed tool, which implements the Beagle graph-based haplotype phasing algorithm, modified to separate the haplotype graph construction and phasing steps. We imputed samples against a single unified imputation reference panel combining the 1000 Genomes Phase 3 haplotypes[70] with the UK10K imputation reference panel[71] using Minimac3[74].

**Association analysis**. In the UK Biobank, we performed GWAS using a linear mixed non-infinitesimal model implemented in BOLT-LMM v2.3[67] across 547,011 genotyped SNPs (minor allele frequency (MAF) ≥ 0.01) and 8,397,536 imputed SNPs (MAF ≥ 0.01 and INFO score ≥0.90), adjusting for genotyping platform and genetic sex. The reference genetic map used was hg19 and linkage disequilibrium scores were generated from European-ancestry individuals taken from the BOLT-LMM package. The genome-wide significance threshold in this analysis was set at $P < 5 \times 10^{-8}$. Conditional regression analysis was performed at the top signal at each of 109 associated loci in BOLT-LMM[67], excluding the MHC region due to the high density of genes and high linkage of variants.

In 23andMe, summary statistics were generated via logistic regression assuming an additive model for allelic effects. Association analysis was performed adjusting for age, sex, the first five principal components, and genotyping platform. The top independent variants from the discovery GWAS were tested for their association with VVs in the 23andMe cohort. Of the 116 genome-wide significant variants in the discovery GWAS, 106 were present in, and passed QC within the replication cohort. The Bonferroni-corrected significance threshold for replication was set at $P < 4.72 \times 10^{-4}$ (0.05/106). Data for all top independent variants that were available in both UK Biobank and 23andMe and that met the SNP QC within 23andMe were meta-analysed using a fixed-effects meta-analysis performed in GWAMA v2.2.2[75].

**Functional annotation of SNPs**. To annotate SNPs at our susceptibility loci, SNP2GENE was performed in Functional Mapping and Annotation of GWAS (FUMA) v1.3.3[20] (https://fuma.ctglab.nl/) using summary statistics from the UK Biobank cohort and default settings. FUMA defined risk loci borders by using all independent genome-wide significant SNPs ($r^2 < 0.6$), and identified all SNPs that were in linkage disequilibrium (LD) with one of these candidate SNPs. Using ANNOVAR (https://annovar.openbioinformatics.org/en/latest/), candidate SNPs within the replicated loci were annotated on the basis of genomic location. Exonic SNPs were investigated further using gnomAD and Ensembl genome browsers (www.ensembl.org/index.html) to identify non-synonymous missense variants (see URLs). All candidate SNPs were annotated with Combined Annotation-Dependent Depletion (CADD)[23] (cadd.gs.washington.edu/), RegulomeDB[24] (www.regulomedb.org/), and 15-core chromatin states (Roadmap ChromHMM model[76]) to predict any regulatory or transcription effects from chromatin states at each SNP.

**Fine-mapping**. For each locus that was replicated in the 23andMe summary statistics ($n = 45$), functionally informed fine-mapping was performed using Polyfun-SuSiE v0.11.92[25] using a 2MB window around each window SNP with the maximum number of causal SNPs set as 5 and a posterior probability threshold of 95%. The LD reference was derived from 1000 Genomes project European population. Fine-mapping was implemented using the echolocatoR package v0.2.2 for R[77].

**Candidate gene mapping**. Four gene mapping approaches—positional mapping, eQTL mapping, MAGMA gene mapping (www.ctg.cncr.nl/software/magma), and summary-based Mendelian randomisation (SMR)—were used to map putative genes at the replicated loci. For FUMA positional mapping, all genome-wide significant SNPs at each locus were mapped to genes within a positional window of 10 Kb[20]. eQTL gene mapping was used to map genes based on having at least one genome-wide significant cis-eQTL in GTEx v8 tibial artery tissue[20]. Using MAGMA v.1.07[27], we conducted a genome-wide, gene-based association study, testing 17,966 protein-coding genes. The significance threshold was set at $P < 2.78 \times 10^{-6}$ (0.05/17966).

To identify gene expression levels associated with VVs due to pleiotropy, SMR and HEIDI analyses were performed using SMR v0.710[28]. Summary statistics from the UK Biobank discovery GWAS were used, alongside eQTL data for GTEx V7 tibial artery (www.gtexportal.org/home/). The top-associated eQTL for each gene was used as an instrumental variable to examine association with VVs. A Bonferroni-corrected significance for all SMR probes was set at $P_{SMR} < 1.01 \times 10^{-5}$

(0.05/4946 probes). Subsequently, a HEIDI test was conducted across all SMR-significant probes to examine for heterogeneity in SMR estimates (significance set at $P < 0.05$/ number of $P_{SMR}$-significant probes).

**Gene set, tissue-specific and pathway enrichment analysis**. Gene-set analysis was implemented in MAGMA v1.07[27] across 15,496 gene sets derived from MSigDB v8.0[78] with a significance threshold of $P < 3.23 \times 10^{-6}$ (0.05/15496) (Supplementary Data 15). Tissue-specific gene property analysis was also performed in MAGMA v1.07 to determine the expression of the protein-coding genes in VV-related tissue types in GTEx v8.0[26].

Pathway analysis was performed in eXploring Genomic Relations (XGR v1.1.3)[29] (https://cran.r-project.org/web/packages/XGR/index.html). XGR ontology enrichment analysis was performed across all mapped genes, in "canonical pathways" with the following settings: hypergeometric distribution testing, any number of genes annotated, any overlap with input genes, and an adjusted FDR < 0.05.

**SNP-based heritability, genetic correlation and PheWAS**. The SNP-based heritability for the UK Biobank cohort was computed using BOLT-REML v2.3, a variance components method[79], and LD score (LDSC) regression using LDSC v1.0.1[19]. LDSC was also used to calculate the LDSC intercept, mean chi-squared test and attenuation score, and SNP-based heritability ($h^2_g$) in the 23andMe cohort[80]. This approach calculates heritability for a specific phenotype by regression of a SNP's association statistic onto its LDSC[80]. LDSC regression was also used to calculate the genetic correlation between VVs and 176 preselected traits, across nine trait categories: metabolites, glycaemic traits, autoimmune diseases, anthropometric traits, smoking behaviour, lipids, cardiometabolic traits, reproductive traits and haematological traits, from publicly available GWAS data within LD Hub (https://ldsc.broadinstitute.org/)[80]. Traits were selected based on putative epidemiological associations with VVs from the literature. We confirmed associations between VVs and the correlated traits at the variant level by employing a PheWAS approach. Each of 49 replicated variants was searched in Open Targets Genetics[31] (https://genetics.opentargets.org/, accessed 25/02/2022), and we collated all phenotypes with $P$-value < 0.005 in the databases available on the platform (UK Biobank, FinnGen, and GWAS Catalog). All phenotypes associated with the 49 variants are shown in Supplementary Data 10.

**Polygenic risk score**. The varicose veins polygenic risk score (VV-PRS) was derived in FinnGen (https://www.finngen.fi/en), a public-private partnership project combining genotype data from Finnish biobanks and digital health record data from Finnish health registries. Genotyping, phenotyping, and quality control of the FinnGen cohort has been reported previously[81], and varicose veins phenotype definition is available at https://risteys.finngen.fi/phenocode/I9_VARICVE.

Summary statistics for varicose vein ICD-10 diagnosis (I9_VARICVE) in FinnGen were downloaded from https://console.cloud.google.com/storage/browser/finngen-public-data-r5/summary_stats/. There were 17,027 VVs cases and 190,028 controls. Summary statistics were lifted over from GRCh38 to GRCh37 using the "default_lift_data" function in the gnomAD package v0.4.0 for Hail v0.2.77 (https://hail.is/). Non-autosomal SNPs were removed and SNP-allele pairs were matched with the provided UK Biobank-derived LD reference comprising high-quality HapMap3 variants using the "snp_match" function in the bigsnpr v1.9.11 package for R. SNPs with an absolute MAF difference >0.1 between FinnGen and UKB were excluded ($n = 36,656$), leaving a total of 1,004,944 SNPs. VVs polygenic risk score was derived using LDpred2[82] (automatic model), following instructions in the package vignette. Model convergence was visually confirmed by inspection of the chain plot. The PLINK2 linear scoring function (–score) was used to calculate the per-sample score for unrelated European participants in UK Biobank ($n = 381,544$).

To validate the predictive utility of this score, we performed logistic regression of varicose vein case/control status (defined as above) against PRS decile, adjusted for year of birth, sex, genotyping array, recruitment centre, body mass index (at recruitment) and PC1-10.

For phenome-wide VV-PRS analysis, Phecode[83]-level time-to-event data was extracted from UKB phenotype data as described previously[84], and varicose vein-specific phenotypes were removed. For each phenotype, Cox regression was performed for phenotype against Z-scored PRS adjusted for year of birth, sex, genotyping array, recruitment centre and PC1-10.

**Bidirectional two-sample Mendelian randomisation analyses**. Genetic instruments were extracted from summary statistics by thresholding at a significance level of $5 \times 10^{-8}$ and clumping according to $r^2 < 0.01$ using a 1000 Genomes European population reference. Varicose vein instruments were extracted from the summary statistics generated above, while other instruments were extracted from the IEU OpenGWAS project[85] (https://gwas.mrcieu.ac.uk/). Exposure and outcome data were harmonised using the "harmonise_data" function in the TwoSampleMR v0.5.6 for R (https://mrcieu.github.io/TwoSampleMR/). The primary analysis performed was the inverse-variance weighted (IVW) method, assuming that all instrumental variables are valid. As sensitivity analyses, we additionally performed

MR-Egger regression to identify available of significant directional pleiotropy. All analyses were implemented in the TwoSampleMR package for R.

**Drug-target enrichment analysis.** The prioritised genes at our replicated loci were queried on the Open Targets Platform[32], assessing whether encoded proteins were tractable to small molecule or antibody targeting, or drug targets in any phase of clinical trial. Genes were also analysed for enriched drug pathways, with a nominal $P < 0.05$ threshold of significance.

**Reporting summary.** Further information on research design is available in the Nature Research Reporting Summary linked to this article.

## Data availability

Discovery GWAS summary statistics from UK Biobank have been deposited in the Oxford University Research Archive and are available for download at: https://doi.org/10.5287/bodleian:8J26woZQg. Full UK Biobank data can be accessed by direct application to UK Biobank. Genotype data for 23andMe research participants have not been deposited in public repositories, as consent for this was not obtained in the study protocol. Summary statistics can be accessed from 23andMe by qualified researchers who enter into agreement with 23andMe that protects subjects' confidentiality. Investigators wishing to collaborate with 23andMe can email dataset-request@23andme.com or apply via the 23andMe research website (www.23andme.com/en-gb/research).

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

## Acknowledgements

The data analysed in the present study was in part provided by the UK Biobank (www.ukbiobank.ac.uk), received under UK Biobank application no. 22572. The replication cohort was provided by 23andMe, Inc. (California). W.A. is supported by the Aziz Foundation, Wolfson Foundation, Royal College Surgeons of England and Oxford NIHR Biomedical Research Centre (Musculoskeletal theme). W.A. was previously supported by grants from the British Association of Plastic and Reconstructive Surgeons (BAPRAS) and Royal College Surgeons of Edinburgh. R.L. is funded by the UK Research and Innovation Future Leaders Fellowship. M.N. and D.F. are supported by the Oxford NIHR Biomedical Research Centre (BMC). A.W. is an NIHR Academic Clinical Lecturer and was supported by an MRC Clinical Research Training Fellowship (MR/N001524/1). The funders had no role in the design of the study; in the collection, analyses or interpretation of data; in the writing of the manuscript or in the decision to publish the results. This represents independent research funded by the NIHR. The view expressed are the authors' own and are not necessarily those of the NIHR, NHS or Department of Health and Social Care.

## Author contributions

D.F, A.W. and K.Z. contributed to the conception, study design, and supervision of this work. W.A., S.K., A.W. and M.N. contributed to data analysis. W.W., A.A. and the 23andMe Research Team contributed to the replication study and analysis. R.L. and A.H. contributed to case ascertainment. W.A., A.W., D.F. prepared the first draft of the manuscript. All co-authors made substantial contributions to data acquisition, data interpretation, and revised the work critically for important intellectual content.

## Competing interests

W.W., A.A. and members of the 23andMe Research Team disclose a relationship with 23andMe, Inc. as employees, and hold stock or stock options in 23andMe, Inc. All other authors have no competing interests to disclose.

## Additional information

**23andMe Research Team**

Michelle Agee[3], Stella Aslibekyan[3], Adam Auton[3], Robert K. Bell[3], Katarzyna Bryc[3], Sarah K. Clark[3], Sarah L. Elson[3], Kipper Fletez-Brant[3], Pierre Fontanillas[3], Nicholas A. Furlotte[3], Pooja M. Gandhi[3], Karl Heilbron[3], Barry Hicks[3], David A. Hinds[3], Karen E. Huber[3], Ethan M. Jewett[3], Yunxuan Jiang[3], Aaron Kleinman[3], Keng-Han Lin[3], Nadia K. Litterman[3], Marie K. Luff[3], Jennifer C. McCreight[3], Matthew H. McIntyre[3], Kimberly F. McManus[3], Joanna L. Mountain[3], Sahar V. Mozaffari[3], Priyanka Nandakumar[3], Elizabeth S. Noblin[3], Carrie A. M. Northover[3], Jared O'Connell[3], Aaron A. Petrakovitz[3], Steven J. Pitts[3], G. David Poznik[3], J. Fah Sathirapongsasuti[3], Anjali J. Shastri[3], Janie F. Shelton[3], Suyash Shringarpure[3], Chao Tian[3], Joyce Y. Tung[3], Robert J. Tunney[3], Vladimir Vacic[3], Xin Wang[3] & Amir S. Zare[3]

