## [Peer Review File · Nature Communications]

Genome-wide association analysis and replication in 810,625 individuals with varicose veinsREVIEWER COMMENTS

Reviewer #1 (Remarks to the Author):

Furniss and colleagues present results of a large GWAS of varicose vein (VV), based on UKB data with replication using 23andMe data. The GWAS is well-conducted, and the authors have been thoughtful in their phenotype definition. The main GWAS is complemented by standard secondary analyses (pathway, gene-based, genetic correlation, etc.). I do have some concerns:

- 1) As individual-level data are available in the UKB, can the authors estimate heritability using one of the many variance component methods (instead of the less precise LDSC approach)?
- 2) Is there any overlap between Mendelian conditions associated with VV and the GWAS loci identified?
- 3) Are the 49 variants identified associated with other traits, whether related to VV or not?
- 4) The discussion is both very long and somehow speculative. At the authors' discretion, I would suggest shortening and summarizing key genes / pathways in a table.
- 5) Have the authors used summary association statistics from the UKB (in combination with 23andMe replication) to derive the wGRS tested in the UKB? This appears to be the case. If so, that would necessarily lead to inflated results, and it needs to be corrected.
- 6) The genetic correlation analysis is interesting but does not inform on what causes VV and what are the consequences of VV. Can the authors confirm the relationships observed by conducting bi-directional Mendelian randomization analyses?

Minor

- 7) In the methods section about GRS, readers are referred to Table 3 for cases definition. However, table 3 is about gene-based enrichment analysis.

Reviewer #2 (Remarks to the Author):

In this manuscript, Ahmed et al perform a genome-wide association study of VV leveraging UK Biobank data for discovery, and 23andme data for independent replication. Through this analysis, they identify 46 associated loci, 29 of which are novel. Downstream analyses using FUMA and other publicly available tools are utilized to prioritize candidate causal genes, and a genetic risk score analysis is proposed. Overall, significant novelty is presented in the current manuscript in the discovery analysis, and the basic tenets of the GWAS (QC, meta-analysis, multiple testing correction) are performed appropriately. However, this reviewer has concerns regarding the current paper:

Major Concerns:

- 1) The genetic risk score analysis methodology is unclear to this reviewer. Generally speaking, GRS analyses are performed by generating weights from one (usually discovery) cohort, and then testing/validating these results in an independent dataset to minimize biases. As currently written, it appears the authors developed a GRS and tested it internally within the datasets used for discovery/GRS creation. If this is accurate, this needs to be addressed/changed, or if an independent dataset was indeed used, this should be clarified.
- 2) Throughout the manuscript, a list of candidate genes/prioritized genes is generated from a series of sources/analyses. While this is interesting at a basic level, this reviewer suggests more complex

analyses be performed with this data. The authors have summary statistics from the largest GWAS of VV to date at their disposal, and the list(s) prioritized genes could further be refined (for example, fine mapping strategies like FOCUS, PMID: 30926970)

3) Along the same lines, the additional variant-level fine mapping or colocalization performed is quite limited. Given the dense imputation panel(s) presented, this data could be leveraged for additional variant level analyses. Leveraging functional annotation data (CADD scores, for example) provides limited novelty to the paper given the large array of publicly available tools that can much more finely describe/map variants

4) With respect to the figures, the figure's main papers do not add much beyond what is already contained within the text. For example, Figures 2,4, and 5 could easily be contained in a supplementary appendix. Their additional value on-top of what is presented in the text is quite limited, and could be enhanced.

5) The use of tibial artery as the tissue of interest for VV eQTL analysis is inappropriate. Varicose veins are a disease process limited to the peripheral, superficial venous system. Tibial artery is an arterial tissue that is affected by atherosclerosis (and to a lesser degree arteritis), completely different disease processes than that which effect the peripheral/superficial veins. This needs to be changed/addressed as the conclusions reached from this data are similarly inappropriate.

Minor concerns:

1) The text on some of the figures exported from FUMA are small and difficult to read. This should be addressed

2) The color scheme representing autoimmune vs anthropometric traits is confusing in the figure. This could be illustrated more easily/clearly

3) The discussion lists a series of genes implicated by the prioritization analyses, but for many of the genes the overall evidence pointing toward a confident, truly causal gene is weak. This should be altered as to not mislead the reader about the confidence of causality demonstrated based on the current data presented.

We are grateful to both reviewers for taking the time to carefully scrutinise our manuscript, and for their many helpful comments and suggestions.

Reviewer #1 (Remarks to the Author):

Furniss and colleagues present results of a large GWAS of varicose vein (VV), based on UKB data with replication using 23andMe data. The GWAS is well-conducted, and the authors have been thoughtful in their phenotype definition. The main GWAS is complemented by standard secondary analyses (pathway, gene-based, genetic correlation, etc.). I do have some concerns:

1) As individual-level data are available in the UKB, can the authors estimate heritability using one of the many variance component methods (instead of the less precise LDSC approach)?

We thank the reviewer for raising this point. We have employed LDSC for our heritability estimates for a number of reasons.

Approaches for estimating variance components typically search for parameter values that maximize the likelihood or the restricted maximum likelihood (REML). Despite a number of algorithmic improvements, computing REML estimates of the variance components on data sets such as the UK Biobank ($\approx 500,000$ individuals genotyped at nearly one million SNPs) remains challenging (Pazokitoroudi et al., Nat Commun. 2020 Aug 11;11(1):4020).

While a number of methods have been proposed to improve the computational efficiency of REML estimators, all of these methods rely on iterative optimization algorithms that do not scale well to biobank-scale datasets consisting of millions of individuals genotyped at tens of millions of SNPs. Further, REML has been shown to yield biased estimates of heritability in ascertained case-control studies (Wu & Sankararaman, Bioinformatics. 2018 Jul 1;34(13):i187-i194).

LDSC is largely robust to confounding due to stratification and shared environmental influences, whereas variance component methods (GREML-SC, -MS, -LDMS, etc.) can give biased estimates insofar as the average LD among SNPs is different from the average LD between SNPs and common variants (Table 1, Evans et al. Nat Genet. 2018 May;50(5):737-745).

LDSC is computationally efficient, easy to implement, and scales well to large datasets. This is probably why the majority of UK Biobank-based genetic association studies, despite having access to individual-level data, have also chosen to employ LDSC rather than variance component methods.

There are a plethora of published methods for estimating SNP-based heritability. Ultimately, each method will have its proponents and detractors. We hope that we have sufficiently justified our use of a well-established and widely adopted method for heritability estimation.

2) Is there any overlap between Mendelian conditions associated with VV and the GWAS loci identified?

We thank the reviewer for raising this interesting point. To answer this question, we have searched the entire OMIM database for Mendelian conditions associated with varicose veins. Only one gene prioritised in our GWAS (*PIEZO1*) has such an association.

Our new Supplementary Table 17 shows all varicose vein-associated Mendelian diseases, and the genes implicated. We reference this in the main text (Discussion, Page 19, with new text underlined):

A previous analysis also suggested FOXC2 to be implicated in the development of varicose veins in the general population⁶⁵, but we did not find evidence of association between varicose veins and FOXC2 in our study. In fact, the only gene identified in this study known to be associated with a Mendelian disease that results in VVs is PIEZO1, which also predisposes to both VVs and lymphoedema (Supplementary Table 17).

3) Are the 49 variants identified associated with other traits, whether related to VV or not?

The only way of comprehensively addressing the reviewer's question would be to generate a large supplementary data file detailing the search results for each of the 49 index SNPs in GWAS Catalog. We would be happy to do this if the reviewer feels strongly that this information would add to the value of the paper.

However, we feel that an altogether more informative approach when it comes to exploring relationships between VVs and other traits is to look at similarities in genetic architecture. In our original submission, we had done this through genetic correlation analyses; in revised manuscript, we have gone further and performed a PheWAS in UK Biobank using a polygenic risk score derived in an independent dataset, with subsequent bidirectional Mendelian randomisation for relevant phenotypes.

This approach has allowed us to make more biologically meaningful interpretations of these signals. For instance, in the Discussion (Page 15), we single out rs3791679, the A allele of which is strongly associated with VV, but also with pelvic organ prolapse and inguinal hernias (one of the phenotypes strongly associated with VV polygenic risk score in the PheWAS). Individuals possessing the A allele at this variant are thus predisposed to diseases characterised by insufficiency in their connective tissues.

4) The discussion is both very long and somehow speculative. At the authors' discretion, I would suggest shortening and summarizing key genes / pathways in a table.

Thank you for this comment. We agree that our discussion was too long and speculative. We have shortened the discussion and considerably narrowed down its focus, dedicating subsections to just three of the biological pathways that garnered the strongest evidence in our analyses (previously five).

The additional analyses that we performed in this revised submission have strengthened the case for a handful of key genes – these now take centre stage in the Discussion, which we believe is now less speculative as a result.

Please note that a summary table of the functional categories of the gene clusters has already been provided (Supplementary Table 16, formerly Supplementary table 12).

5) Have the authors used summary association statistics from the UKB (in combination with 23andMe replication) to derive the wGRS tested in the UKB? This appears to be the case. If so, that would necessarily lead to inflated results, and it needs to be corrected.

Both Reviewers 1 and 2 helpfully pointed out problems with the weighted genetic risk score analyses in our original manuscript. We therefore made the decision to discard these analyses, and have adopted an entirely different approach in our construction of a genetic risk score.

We have now constructed a conventional polygenic risk score (PRS) using the LDpred2 method in an independent sample of VVs cases and controls in FinnGen. We have validated this in UK Biobank, demonstrating a strong correlation between PRS decile and VVs case/control status (Results, Page 9; Figure 3; Supplementary Table 9). We further demonstrate that this PRS has predictive utility by virtue of its correlation with a more severe VV phenotype (Results, Page 9-10). We then use this FinnGen-derived PRS to perform a phenome-wide association study in UK Biobank (Results, Page 10; Supplementary Table 11), and perform bidirectional Mendelian randomisation to establish causal relationships for relevant phenotypes (Results, Page 11; Supplementary Table 12).

We believe that these extensive PRS analyses have considerably enhanced our paper, and thank the reviewers for prompting these analyses.

6) The genetic correlation analysis is interesting but does not inform on what causes VV and what are the consequences of VV. Can the authors confirm the relationships observed by conducting bi-directional Mendelian randomization analyses?

This has now been done. Please see the answer to point 5, above.

7) In the methods section about GRS, readers are referred to Table 3 for cases definition. However, table 3 is about gene-based enrichment analysis.

Thank you for pointing out this error. Given that we have completely re-done our genetic risk score analyses, this table no longer exists. We have ensured that all figures and tables referenced in the revised manuscript are correctly labelled.

Reviewer #2 (Remarks to the Author):

In this manuscript, Ahmed et al perform a genome-wide association study of VV leveraging UK Biobank data for discovery, and 23andme data for independent replication. Through this analysis, they identify 46 associated loci, 29 of which are novel. Downstream analyses using FUMA and other publicly available tools are utilized to prioritize candidate causal genes, and a genetic risk score analysis is proposed. Overall, significant novelty is presented in the current manuscript in the discovery analysis, and the basic tenets of the GWAS (QC, meta-analysis, multiple testing correction) are performed appropriately. However, this reviewer has concerns regarding the current paper:

Major Concerns:

1) The genetic risk score analysis methodology is unclear to this reviewer. Generally speaking, GRS analyses are performed by generating weights from one (usually discovery) cohort, and then testing/validating these results in an independent dataset to minimize biases. As currently written, it appears the authors of developed a GRS and tested it internally within the datasets used for discovery/GRS creation. If this is accurate, this needs to be addressed/changed, or if an independent dataset was indeed used, this should be clarified.

Both Reviewers 1 and 2 helpfully pointed out problems with the weighted genetic risk score analyses in our original manuscript. We therefore made the decision to discard these analyses, and have adopted an entirely different approach in our construction of a genetic risk score.

We have now constructed a conventional polygenic risk score (PRS) using the LDpred2 method in an independent sample of VVs cases and controls in FinnGen. We have validated this in UK Biobank, demonstrating a strong correlation between PRS decile and VVs case/control status (Results, Page 9; Figure 3; Supplementary Table 9). We further demonstrate that this PRS has predictive utility by virtue of its correlation with a more severe VV phenotype (Results, Page 9-10). We then use this FinnGen-derived PRS to perform a phenome-wide association study in UK Biobank (Results, Page 10; Supplementary Table 11), and perform bidirectional Mendelian randomisation to establish causal relationships for relevant phenotypes (Results, Page 11; Supplementary Table 12).

We believe that these extensive PRS analyses have considerably enhanced our paper, and thank the reviewers for prompting these analyses.

2) Throughout the manuscript, a list of candidate genes/prioritized genes is generated from a series of sources/analyses. While this is interesting at a basic level, this reviewer suggests more complex analyses be performed with this data. The authors have summary statistics from the largest GWAS of VV to date at their disposal, and the list(s) prioritized genes could further be refined (for example, fine mapping strategies like FOCUS, PMID: 30926970)

Please see our response to point 3, below.

3) Along the same lines, the additional variant-level fine mapping or colocalization performed is quite limited. Given the dense imputation panel(s) presented, this data could be leveraged for additional variant level analyses. Leveraging functional annotation data (CADD scores, for

example) provides limited novelty to the paper given the large array of publicly available tools that can much more finely describe/map variants¹

We thank the reviewer for the comments and suggestions raised here, and in point (2), above. We agree in the value of performing more sophisticated gene mapping than we had done in our original submission.

There are a myriad different ways of mapping GWAS hits to genes. We have chosen to use the well-established technique of functionally-informed fine-mapping using Polyfun-SuSiE (Weissbrod et al. Nat Genet. 2020 Dec;52(12):1355-1363).

We identify variants with a posterior probability >95% at our replicated loci, and incorporate several of these variants in the context of their respective genes in the Discussion.

4) With respect to the figures, the figure's main papers do not add much beyond what is already contained within the text. For example, Figures 2,4, and 5 could easily be contained in a supplementary appendix. Their additional value on-top of what is presented in the text is quite limited, and could be enhanced.

We thank the reviewer for pointing out the figures that are redundant – we agree. We have moved the original Figure 4 (MAGMA tissue expression analysis) to the Supplementary appendix, and did the same for Figure 3 (functional annotation of significant SNPs).

Re: the original Figure 2 (QQ plot + Manhattan), we agree that panel A (QQ plot) is redundant; however, we feel that panel B (colour-coded Manhattan with locus names) is quite helpful in distinguishing discovery/replication/novel/non-novel loci. It also allows the reader to get an intuitive and visual appreciation of the strength of certain signals over others. We hope that the reviewer finds it acceptable that panel A has been moved to the Supplementary as its own figure, while panel B has been retained in the main manuscript as a figure in its own right. It has been considerably enlarged, and is now easier to read.

Re: the original Figure 5 (graphical representation of genetic correlations with other phenotypes), we realise that this adds nothing beyond what was shown in the original Supplementary Table 8 (now Supplementary Table 10). As such, we have removed this figure entirely.

5) The use of tibial artery as the tissue of interest for VV eQTL analysis is inappropriate. Varicose veins are a disease process limited to the peripheral, superficial venous system. Tibial artery is an arterial tissue that is affected by atherosclerosis (and to a lesser degree arteritis), completely different disease processes than that which effect the peripheral/superficial veins. This needs to be changed/addressed as the conclusions reached from this data are similarly inappropriate.

We thank the reviewer for this comment. While GTEx is used widely as a useful resource for tissue-specific gene expression, one of its principal limitations is that it includes only a relatively small proportion of all human tissue types.

Embryologically, arterial and venous tissues develop through the shared pathway of vasculogenesis. The two vessel types share a common basic cellular architecture of endothelium/tunica media/tunica adventitia. Of the 53 cell and tissue types available in the GTEx resource, tibial artery is the closest to lower limb venous tissue. While acknowledging the considerable limitations in using arterial tissue as a surrogate for venous tissue, we feel strongly that this is preferable to using eQTL data from e.g. whole blood.

We have encountered similar dilemmas in the past. For example, in our previous paper on carpal tunnel syndrome published in this journal (Wiberg et al, Nat Commun. 2019 Mar 4;10(1):1030), we used GTEx fibroblast eQTL data as a surrogate for the connective tissues surrounding human tendons, on the basis that fibroblasts are the principal cellular component of this tissue of interest. We do not think it unreasonable to perform a similar extrapolation from arterial to venous tissue in this case, as it is anatomically, embryologically, and histologically the closest tissue.

Minor concerns:

1) The text on some of the figures exported from FUMA are small and difficult to read.

As per our response to point 4, above, we hope that we have addressed this issue by moving most of our figures from the main text to the Supplementary appendix, where they have been enlarged and presented in full-size.

2) The color scheme representing autoimmune vs anthropometric traits is confusing in the figure. This could be illustrated more easily/clearly

As per our response to point 4, above, this figure has now been removed entirely, and the data can be found in Supplementary Table 10.

3) The discussion lists a series of genes implicated by the prioritization analyses, but for many of the genes the overall evidence pointing toward a confident, truly causal gene is weak. This should be altered as to not mislead the reader about the confidence of causality demonstrated based on the current data presented.

We thank the reviewer for highlighting this. We believe that the functionally-informed fine-mapping that we have performed for this revised manuscript has strengthened the causal SNP->gene inference for the most important genes. We have greater confidence in these genes, and they have accordingly been given a greater focus in the Discussion. At the same time, we have made substantial cuts to our previous discussions centred around the more speculative genes.

We agree with the Reviewer's sentiments about the need to be cautious so as not to mislead the reader about the confidence of causality demonstrated. We have therefore included the following sentences in the Discussion (Page 20) to acknowledge this as a limitation:

"Finally, we acknowledge that out of the many VVs-associated genes identified by a multitude of gene-prioritisation and mapping methods, only a small minority have robust evidence linking them to VVs through a truly causal variant. Functional validation will be required in diseased venous tissue to confidently associate these candidate susceptibility genes to VVs."

REVIEWER COMMENTS

Reviewer #1 (Remarks to the Author):

I would like to thank the authors for responding to my prior comments. However, I do have some concerns:

- Regarding my prior comment #1. Many methods, such as BOLT, have been developed specifically for use in biobank-scale datasets. Can the authors estimate heritability using BOLT? There is no consensus on the best method and showing consistent results with two different approaches is important. I am also confused about the comment about ascertained case-control study design as the UKB was not ascertained for disease status. In other words, varicose vein cases were not oversampled relative to the disease prevalence.

- Regarding my prior comment #3. Can the authors report and comment on significant associations between the 49 variants identified and other traits? This could be done using a PheWAS approach in the UKB, FinnGen or other large database (such as GWAS Catalogue as mentioned by authors). Only significant associations need to be reported.

Reviewer #2 (Remarks to the Author):

In the revised manuscript, the authors make a series of changes that substantially improve the manuscript. Overall, the authors address nearly all of this reviewer's concerns.

This reviewer does have one lingering concern. While the authors, in their reviewer responses, do present some reasonable arguments for the use of tibial artery eQTL data for this manuscript, this reviewer still believes this has substantial methodologic limitations. However, if the authors present their reasoning for using arterial tissue for this purpose, and mention this in the paper's limitation, this would satisfy all of this reviewer's concerns.

We are grateful to both reviewers for taking the time to carefully scrutinise our revised manuscript. We believe the remaining points have now been adequately addressed, and hope the additional analyses and changes will satisfy the reviewers.

Reviewer #1 (Remarks to the Author):

I would like to thank the authors for responding to my prior comments. However, I do have some concerns:

- Regarding my prior comment #1. Many methods, such as BOLT, have been developed specifically for use in biobank-scale datasets. Can the authors estimate heritability using BOLT? There is no consensus on the best method and showing consistent results with two different approaches is important. I am also confused about the comment about ascertained case-control study design as the UKB was not ascertained for disease status. In other words, varicose vein cases were not oversampled relative to the disease prevalence.

Thank you for recommending the use of BOLT to derive the heritability. We have now computed the SNP-based heritability of varicose veins in the UK Biobank cohort using the variance components method in BOLT-REML.

The heritability value has been updated in the main manuscript (page 5):

We estimated the total SNP heritability (h^2_g) for VVs in UK Biobank to be 8.04% (S.E. = 0.17%).

And we refer to this in the Methods (page 38):

The SNP-based heritability for the UK Biobank cohort was computed using BOLT-REML, a variance components method.⁸⁰

Reference 80: Loh P-R et al. Nat Genet. 2015 Dec;47(12):1385-92. doi: 10.1038/ng.3431

- Regarding my prior comment #3. Can the authors report and comment on significant associations between the 49 variants identified and other traits? This could be done using a PheWAS approach in the UKB, FinnGen or other large database (such as GWAS Catalogue as mentioned by authors). Only significant associations need to be reported.

We are grateful to the reviewer for this recommendation. As suggested, we have taken a PheWAS approach to look for associations between our 49 replicated SNPs and phenotypes in UKB, FinnGen, and GWAS Catalog, using the Open Targets Genetics platform.

This uncovered the association between three of our lead variants with systemic lupus erythematosus (SLE), cementing the finding from our LDSC genetic correlation analysis at the SNP level.

The full PheWAS data are presented as Supplementary Data 3, and we have appended the following text to the document to incorporate this new analysis.

Results (page 10):

*We used the Open Targets Genetics platform³² to confirm the associations between VVs and SLE at the variant level, using a phenome-wide association study (PheWAS) approach (**Supplementary Data 3**); three of our 49 replicated lead variants (rs4849044, rs7773004, rs61863928) demonstrated an association with SLE.*

Methods (page 39):

*We confirmed associations between VVs and the correlated traits at the variant level by employing a PheWAS approach. Each of 49 replicated variants was searched in Open Targets Genetics³² (accessed 25/02/2022), and we collated all phenotypes with P -value < 0.005 in the databases available on the platform (UK Biobank, FinnGen, and GWAS Catalog). All phenotypes associated with the 49 variants are shown in **Supplementary Data 3**.*

*Reference 32: Ghossaini, M. et al. Open Targets Genetics: Systematic identification of trait-associated genes using large-scale genetics and functional genomics. *Nucleic Acids Res.* **49**, D1311–D1320 (2021).*

Reviewer #2 (Remarks to the Author):

In the revised manuscript, the authors make a series of changes that substantially improve the manuscript. Overall, the authors address nearly all of this reviewer's concerns.

This reviewer does have one lingering concern. While the authors, in their reviewer responses, do present some reasonable arguments for the use of tibial artery eQTL data for this manuscript, this reviewer still believes this has substantial methodologic limitations. However, if the authors present their reasoning for using arterial tissue for this purpose, and mention this in the paper's limitation, this would satisfy all of this reviewer's concerns.

We appreciate the Reviewer's concern regarding the use of tibial artery eQTL. We now explicitly acknowledge the limitation of using tibial artery tissue data to draw inferences on lower limb venous tissue. The new text has been underlined.

Discussion (page 20-21):

Finally, we acknowledge that out of the many VVs-associated genes identified by a multitude of gene-prioritisation and mapping methods, only a small minority have robust evidence linking them to VVs through a truly causal variant. Moreover, our eQTL-based gene mapping and summary-based Mendelian randomisation analyses were limited by our lack of access to eQTL data for venous tissue. We therefore had to employ GTEx tibial artery tissue data as an imperfect surrogate, reasoning that it is the anatomically, histologically, and embryologically closest tissue to lower limb veins within the GTEx dataset. Functional validation will be required in diseased venous tissue to confidently associate these candidate susceptibility variants and genes to VVs.